# Participatory Systems Modelling for Youth Mental Health: An Evaluation Study Applying a Comprehensive Multi-Scale Framework

**DOI:** 10.3390/ijerph19074015

**Published:** 2022-03-28

**Authors:** Grace Yeeun Lee, Ian Bernard Hickie, Jo-An Occhipinti, Yun Ju Christine Song, Salvador Camacho, Adam Skinner, Kenny Lawson, Samuel J. Hockey, Adriane Martin Hilber, Louise Freebairn

**Affiliations:** 1Brain and Mind Centre, The University of Sydney, Sydney, NSW 2050, Australia; ian.hickie@sydney.edu.au (I.B.H.); jo-an.occhipinti@sydney.edu.au (J.-A.O.); yun.song@sydney.edu.au (Y.J.C.S.); adam.skinner@sydney.edu.au (A.S.); kenny.lawson@sydney.edu.au (K.L.); samuel.hockey@sydney.edu.au (S.J.H.); louise.freebairn@sydney.edu.au (L.F.); 2Computer Simulation & Advanced Research Technologies (CSART), Sydney, NSW 2021, Australia; 3Swiss Centre for International Health, Swiss Tropical and Public Health Institute, 4123 Allschwil, Switzerland; salvador.camacho@swisstph.ch (S.C.); adriane.martinhilber@swisstph.ch (A.M.H.); 4University of Basel, 4001 Basel, Switzerland; 5Research School of Population Health, The Australian National University, Canberra, ACT 0200, Australia

**Keywords:** youth mental health, policy, participatory modelling, participatory systems modelling, evaluation framework, evaluation criteria, systems modelling and simulation, strategic decision making, stakeholder-based modelling, study protocol

## Abstract

The youth mental health sector is persistently challenged by issues such as service fragmentation and inefficient resource allocation. Systems modelling and simulation, particularly utilizing participatory approaches, is offering promise in supporting evidence-informed decision making with limited resources by testing alternative strategies in safe virtual environments before implementing them in the real world. However, improved evaluation efforts are needed to understand the critical elements involved in and to improve methods for implementing participatory modelling for youth mental health system and service delivery. An evaluation protocol is described to evaluate the feasibility, value, impact, and sustainability of participatory systems modelling in delivering advanced decision support capabilities for youth mental health. This study applies a comprehensive multi-scale evaluation framework, drawing on participatory action research principles as well as formative, summative, process, and outcome evaluation techniques. Novel data collection procedures are presented, including online surveys that incorporate gamification to enable social network analysis and patient journey mapping. The evaluation approach also explores the experiences of diverse stakeholders, including young people with lived (or living) experience of mental illness. Social and technical opportunities will be uncovered, as well as challenges implementing these interdisciplinary methods in complex settings to improve youth mental health policy, planning, and outcomes. This study protocol can also be adapted for broader international applications, disciplines, and contexts.

## 1. Introduction

Health systems internationally are under immense pressure to manage complex and compounding health challenges, often being met with policy resistance [1,2]. The introduction of new policies is often top-down and as such can be opposed by health service providers and consumers [3]. This is also true in mental health, which is currently one of the top public health concerns that affects all aspects of individual lives, as well as the health and wellbeing of families and whole communities [4,5,6]. In particular, young people are the most vulnerable, as mental disorders and substance misuse affect at least 1 in 4 young people by the time they reach 25 years of age and are the leading causes of disability and premature death in this age group globally [7]. Mental disorders and substance abuse that emerge during adolescence can persist into later adulthood, having lifelong implications [8,9].

Despite Australia being one of the first countries to develop and implement a national mental health plan in 1993 [10], its mental health system continues to be underdeveloped and over-stretched, faced with persistent challenges including but not limited to service fragmentation, late intervention, mental health treatment isolated from other physical and social needs, poor quality of care, and inefficient resource allocation [11,12,13]. This has led to attempts by the government to invest billions of dollars in mental health system reforms, programs, and services, which has achieved little impact in the population’s mental health outcomes [14]. Insufficient evaluations of such mental health system reforms, programs, and services contribute to an ongoing cycle of refunding and reactive acts of government without truly understanding which initiatives would yield the best returns on investment [15].

### 1.1. Supporting Strategic Decision Making with Participatory Systems Modelling

The deployment of systems models offers a valuable tool to support strategic decision making for some of the world’s most complex challenges, including the current global mental health crisis that is particularly exacerbated in young people. Systems models consider the complex causal inter-relationships that drive population mental health outcomes and can be used to simulate policies and initiatives to determine which combinations are likely to deliver the best outcomes and returns on investment in youth mental health [2]. In particular, system dynamics modelling is a well-established quantitative method long used in other sectors [16]. System dynamics modelling can also lend itself to a participatory approach whereby an interdisciplinary stakeholder group is actively engaged in mapping and contextualizing causal mechanisms driving complex system behaviors [17]. This process is commonly referred to as participatory systems modelling (PSM). The involvement of stakeholders in the model building process is particularly significant in mental health, as it offers the opportunity for the perspectives of community representatives from across the system—including young people with lived (or living) experience of mental illness—to contribute to the technical model development process [18]. This enables improved model credibility, utility, and a robust basis for policy and planning dialogues which can lead to organized, evidence-informed, and active advocacy [16].

Systems models also have the ability to incorporate local contextual factors [19,20]. Such factors include but are not limited to local service capacity, quality, and affordability that contribute to delays in service access, incentives and disincentives in the system, as well as other characteristics and locally specific contextual drivers of mental illness and suicidal behavior such as high youth unemployment, a lack of education and training opportunities, and intergenerational disadvantages [17,21,22].

### 1.2. The Need to Evaluate Participatory Systems Modelling Programs

Though PSM methods have been widely adopted across disciplines, including in the business and environmental sectors [16,17,23,24], they are only more recently being applied to support youth mental health systems planning and policy [17,25]. PSM in this area offers much promise in improving understanding and ownership of systems issues among the diverse stakeholders involved, providing an opportunity to explore the implications of alternate resource allocation strategies or policies, improving communication and coordination across the system, building consensus and collaborative action, and delivering transparency in decision making [26,27,28]. However, there is need for improved evaluation efforts in PSM programs to understand their value in different contexts [26].

Reflective evaluation practices that include feedback from system stakeholders enables an iterative process that investigates whether the participatory modelling program has achieved its aims. Such aims may include but are not limited to building stakeholder capacity in understanding the behavior of complex systems, forming connections between stakeholders from different parts of the system to improve coordination (and reduce fragmentation), fostering confidence in using the systems model, accountability to better inform decision making, as well as consensus for collaborative action to improve the system at the national, state, and local government levels [26].

A scoping review of participatory modelling evaluation frameworks is described elsewhere. This review provides a synthesis of the literature and presents a comprehensive multi-scale evaluation framework that can be applied to assess participatory modelling programs across diverse disciplines as well as across different modelling methods. The framework aims to assess four key outcomes: *feasibility* (i.e., is participatory modelling feasible?), *value* (i.e., what is the value of the participatory modelling process?), *impact* (i.e., what changed or was actioned as a result of the participatory modelling process?), and *sustainability* (i.e., are the changes and actions from the participatory modelling process sustained over time?). To our knowledge, this evaluation study protocol is the first of its kind, applying a novel evaluation framework and techniques in the context of PSM for youth mental health research.

### 1.3. Right Care, First Time, Where You Live Program

A team of multidisciplinary mental health researchers at The University of Sydney in Australia has been developing system dynamics models for youth mental health utilizing a participatory approach [17,18,19,25]. *Right care*, *first time*, *where you live* is a national multi-site health services research program (henceforth referred to as the Program) that will work with eight sites across Australia, delineated by local health regions in the Australian Capital Territory, New South Wales, Queensland, and Western Australia. As part of the Program, system dynamics models will be co-designed and implemented at each of the eight participating sites. A national Australian systems model will also be developed to ensure the expansion of the modelling infrastructure can be applied to youth mental health systems across the country through evidence-informed decision making in mental health policy and planning [29]. Blueprints describing the development of the technical systems models [30], the participatory modelling approach [31], as well as the overall protocol for the Program are provided elsewhere.

This research protocol describes the PSM evaluation of the Program. The feasibility and value of PSM to support decision making have previously been evaluated for population health issues including obesity prevention and diabetes in pregnancy [32]. However, important gaps in knowledge remain, such as evaluating the PSM approach for youth mental health policy and planning, system strengthening, and the inclusion of the perspectives of diverse stakeholders including young people with lived (or living) experience as well as their carers in the participatory (co-design) process. This evaluation study will contribute new knowledge by comprehensively analyzing how developing and implementing co-designed systems models can influence decision making, stakeholder engagement and learning, consensus, as well as collaboration to positively impact the youth mental health landscape.

### 1.4. Objectives

This study applies a comprehensive multi-scale evaluation framework designed for participatory modelling programs that is described elsewhere and aims to answer four research questions:Is it feasible to undertake an inclusive, transparent, participatory approach to develop highly technical and broadly scoped youth mental health systems models in a way that local stakeholders can understand (i.e., the structure, logic and assumptions of the systems model) and find credible?Does the co-designed systems model add value to decision making for youth mental health system strengthening or resource allocation?What has been impacted as a result of the PSM process (i.e., what changes or actions has the PSM process facilitated)?Have these impacts (changes or actions) been sustained over time to improve youth mental health outcomes?

The experiences arising from the application of PSM among diverse stakeholder groups will be explored, including but not limited to the following:Health organization administrators, including staff from funding agencies;Front-line health service professionals, including clinicians;Community representatives, including educators and young people with lived (or living) experience of mental illness.

## 2. Materials and Methods

The described evaluation study protocol was reviewed and approved by the Sydney Local Health District Human Research Ethics Committee (HREC; Protocol No X21-0151 and 2021/ETH00553) on 5 July 2021.

### 2.1. Study Design and Setting

This protocol describes an evaluation study of the PSM process of the Program that applies a comprehensive multi-scale evaluation framework drawing on participatory action research (PAR) principles and methods described elsewhere. This evaluation study also adopts formative, summative, process, and outcome evaluation techniques. The PAR approach of action and reflection enables the flexibility to iterate the evaluation study throughout the Program, responding to different research contexts as needed for the eight participating sites [33]. A co-design process involving three PSM workshops with local stakeholders will be conducted at each participating site to develop eight local youth mental health systems models. The evaluation process will be aligned to the PSM workshops and will be conducted in two sites per year from 2022 to 2025. The evaluation approach, including the data collection procedures, are described below. Figure 1 summarizes the time points of evaluation at each site aligned to the PSM workshops.

### 2.2. Study Population

To best capture the Australian mental health context, the eight research sites will vary across urban, outer-urban, regional, and rural-remote locations and may include diverse cultural or socioeconomic subgroups [18]. Up to 55 participants will be included per site (8 sites in total, *N* ≈ 440 participants). The proposed sample size is based on the extensive experience of the research team in conducting PSM studies [2,18,32], and reflects the likely number of participants needed to capture diverse perspectives across the system and achieve the saturation of themes [34].

### 2.3. Inclusion and Exclusion Criteria

Diverse stakeholders from each local youth mental health and broader social systems will be invited to participate in the PSM workshops. Invited representatives will include but are not limited to young people with lived (or living) experience, supportive others (e.g., family members, carers, and close friends), health professionals, service managers, policy makers, local academics, educators (e.g., teachers), and administrators.


Inclusion Criteria:


Individuals (≥14 years), supportive others (e.g., carers), as well as professionals who work with young people or are involved in the local youth mental health and broader social systems;English proficiency;Capacity to give written informed consent and willingness to participate in the study.

### 2.4. Recruitment Procedure

The recruitment process for this study will be undertaken in collaboration with the primary partner organization at each of the participating sites with guidance from the research team. Active snowball sampling will be employed when appropriate to enable PAR, whereby the invited participants are empowered to contribute and identify any other stakeholders who should also be invited to participate. Throughout this process, a local chief investigator for each participating site will be identified to play a critical role in both recruiting key stakeholders as well as facilitating the engagement of local communities [35,36]. Eligible participants will be enrolled into the study after providing informed consent through a transparent communication process to ensure that all stakeholders can make their own decisions to participate [37]. The privacy of all participants will be emphasized, and all potential participants will be informed that their data will be non-identifiable and securely stored [38,39].

As populations from culturally and linguistically diverse backgrounds, including Aboriginal and Torres Strait Islander communities, may be invited to participate across the eight sites, recruitment will be conducted in an ethical and effective manner to ensure engagement in intercultural settings will be respectful, reflective, and reasoned [40,41,42]. Cultural competency in research is the ability to provide high-quality research that considers the culture and diversity of populations when developing ideas, conducting research, and exploring applicability of research findings [43].

### 2.5. Data Collection Procedures

A mixed methods approach will be adopted, as this strengthens the study design through the triangulation, cross-verification, and validation of evaluation data [44]. Data collection will occur via online surveys (which will incorporate patient journey mapping and social network analysis), semi-structured interviews, researcher observations and recordings from the PSM workshops, meetings with local stakeholders outside the PSM workshops, reflections and field notes made by the research team throughout the Program, and monitoring stakeholder use of the final systems model by configuring the model interface to collect user tracking data by page. Participants will be reimbursed for their time participating in online surveys and semi-structured interviews.

Table 1 summarizes the mixed method application of the comprehensive multi-scale evaluation framework. Table 1 was utilized when developing the interview and online survey questions to ensure that all evaluation criteria of the comprehensive multi-scale framework were incorporated into the data collection procedures. Data may also be compared across (e.g., rural vs. urban) and within (e.g., differing experiences across subgroups of participants) the participating sites to analyze how the feasibility, value, impact, and sustainability of PSM varies depending on the implementation context. Cross-site and subgroup analyses will likely lead to consideration of the impact of broader determinants that may affect participation in and outcomes of the PSM process [8,9,10,12].
Online Surveys

Surveys will be conducted with local stakeholders participating in the Program for each site at three time points. Quantitative and qualitative data will be collected (1) prior to commencement of the PSM workshop series (*baseline*), (2) on completion of the PSM workshop series and systems model development (*first follow-up*: around month 6), and (3) 6 months following the completion of the PSM workshop series (*final follow-up*: around month 12). The surveys, adapted from Freebairn et al. [34], have four main components: (1) questions to elicit perceptions regarding priorities for youth mental health system strengthening, (2) expectations and experiences of participating in the PSM process, (3) social network analysis, and (4) patient journey mapping.

All local stakeholders invited to participate in the PSM workshops will be invited to complete online surveys. The survey questions have undergone cognitive testing to ensure they are relevant, logical, and easily understood by participants [45]. Specifically, the survey questions have been reviewed by the Brain and Mind Centre’s Youth Lived Experience Working Group to ensure that the questions are appropriate and suitable for young people with lived (or living) experience of mental illness. Numerous changes were made from the working group’s feedback, including rephrasing the questions to utilize strength-based language. Surveys will be administered through a secure online data collection tool, Cogniss, which is an online platform that has the ability to gamify survey components [46]. Gamification of online surveys has been acknowledged to improve participant engagement, reduce bias, and improve the quality of survey results [47]. The online surveys will incorporate game design elements such as avatars and badges to motivate respondents to complete activities. Gamification techniques such as competition with others will not be utilized to ensure that surveys remain non-identifiable. The process of prototyping, developing, and iterating the gamified activities through usability testing is described elsewhere [48]. Usability testing is distinct from cognitive testing, as the focus will be on the usability of the online survey (e.g., how well the gamified activities are received by a diverse sample to represent stakeholders that may attend the PSM workshops) [49].

Within the online surveys, two activities will be gamified, specifically social network analysis and patient journey mapping. Social network analysis is a methodology to quantitatively measure and visually analyze social relationships and how they change over time [50,51]. As online surveys will be configured so that questions can be personalized to the participant depending on how they have identified themselves (e.g., young person vs. professional), only the participants who have identified as working in a professional capacity in the youth mental health or broader social system will be asked to complete questions related to social networks, with the aim to measure if interdisciplinary collaboration has changed (i.e., improved or been made worse) as a result of the PSM process. The customization of survey questions to particular demographic groups ensures that the questions are relevant and minimizes respondent confusion or fatigue [52]. A preview of the gamified social network analysis activity is shown in Figure 2.

To understand how young people navigate the local youth mental health system at each participating site—and to uncover the potential strengths and challenges of the system—a patient journey mapping activity will be included within the online surveys. All participants will be presented with up to two vignettes that describe young people who are seeking mental health care. Using these two case studies, respondents will be asked to participate in a ‘complete the story’ activity and map out how a typical journey might look if the young person were to seek help and navigate their community’s youth mental health system. Participants can also optionally share their own personal experiences of navigating their community’s mental health system. Importantly, respondents will have the option to freely choose which story they would like to explore (i.e., fictional case study vs. personal experience). The patient journey mapping activity is followed by additional survey questions regarding the current gaps in the community’s mental health services, such as waiting times to make an appointment with the local youth mental health care service(s). The patient journey mapping activity will be included for both the baseline and follow-up surveys to understand if participants’ contributions to the PSM workshops led to any changes in perceived knowledge, beliefs, assumptions, or engagement behaviors with the local mental health system. Figure 3 shows a preview of the patient journey mapping activity.

All participants will also be asked to complete a series of questions within the online surveys to rate the importance of youth mental health interventions. Through these questions, data will be analyzed to understand the similarities and differences among participants across the system before and after participating in the PSM workshops (e.g., measuring any convergence toward consensus) to specifically understand how the ratings of importance become more targeted and in line with what the final systems model is indicating are the best interventions options (i.e., measuring systems learning).

Box 1 provides example questions from the online survey. The complete baseline and follow-up online surveys have been provided as Appendix A.


Semi-Structured Interviews


Participants invited to the PSM workshops will be asked to participate in one-on-one interviews using questions adapted from Freebairn et al. [2] aligned to the time points for the online surveys (i.e., 1 month before the first PSM workshop, immediately after the last PSM workshop, and 6 months following the last PSM workshop). Interviews will take up to 1 h and conducted either face to face or via appropriate telecommunication tools (such as videoconferencing). Semi-structured interviews will be conducted to understand any detected change (or lack thereof) in participant perspectives regarding the current challenges and priorities of youth mental health services in their local communities, their experiences with and roles within their local community’s youth mental health and broader social systems, their previous experiences co-designing systems models, their experience of participating in this research Program, and their experience using (or their experience contributing to the development of) the systems model as a decision support tool in their community. Recruitment will continue until thematic saturation is achieved across the participant subgroups. Qualitative data collected from the interviews will be cross-verified with the quantitative data collected via the online surveys as a means to triangulate and validate data, forming a richer understanding of participants’ experience engaged in the PSM process. For example, questions regarding interdisciplinary collaboration probed during the interviews will be used to cross-check data from the online survey responses to conduct the social network analysis. An excerpt example of the semi-structured interview questions is provided in Box 2. The full baseline and follow-up semi-structured interview guides are provided as Appendix A.

Box 1Example questions from the online survey.
**Topic: Expectations for the research Program**

** Questions in this topic elicit understanding of the ‘Value’ category of the comprehensive multi-scale evaluation framework.*
What do you hope will be the benefits of the systems model ‘what if’ tool for your community? Please select all that apply:
☐Improve the health and wellbeing of young people in my community☐Assist in making better decisions to improve youth mental health treatments or programs ☐Address the current gaps in my community’s youth mental health system
○*If this option is ticked*: What do you think are the current gaps that exist in your community’s youth mental health system (e.g., cannot get an appointment because of long wait lists)? 
☐Improve my understanding of the youth mental health system in my community☐Better navigate the mental health services or organizations in my community (e.g., to find the right care, to make correct referrals to other services, etc)☐Other: 


Box 2Example questions used during a semi-structured interview.
**Topic: Current youth mental health system**

** Questions in this topic elicit understanding of the ‘Change and Action (Impact)’ and ‘Sustainability’ categories of the comprehensive multi-scale evaluation framework.*

Based on your experience, what are the current challenges of youth mental health care in your community?What do you think is driving these challenges? What changes do you think are required to improve youth mental health care?Why do you think these changes have not yet been implemented?


### 2.6. Data Analysis Plan

***Online survey analysis:*** Statistical analysis methods such as multilevel regression modelling will be used to assess the impact of time (baseline, first follow-up, and final follow-up) on the dependent variables, including participants’ attitudes regarding the feasibility of the PSM process, participants’ attitudes regarding the value of the PSM process, and changes in participants’ attitudes or actions (e.g., improved consensus of participants) due to the PSM process.

***One-on-one interview analysis:*** A thematic analysis of qualitative data will be conducted to identify patterns or themes, guided by the approach described by Braun and Clarke [53,54]. An iterative process of descriptive coding and analytical memos will also be adopted to identify themes and categories [55]. Interview data will be coded and cross-checked by a senior researcher to assess interrater reliability, which will also be iteratively reviewed by senior members of the research team. Relevant computer software may be used, such as professional transcription tools (e.g., Rev Speech-to-Text), in addition to standard Microsoft packages including Excel and Word.

***Social network analysis:*** Qualitative and quantitative data collected through online surveys and semi-structured interviews will be considered when conducting the social network analysis. Researcher observations, field notes, as well as Program outputs from the PSM process may also be used to further validate and contextualize the social network analysis. Visual analysis or mapping the identified social networks via algorithm-generated network diagrams will be completed using appropriate computer software packages such as Gephi [56]. Through this analysis, several ways to examine social connections can be considered, such as connectedness (number of connections one node—or actor that makes up the network—has to other nodes), reciprocity (level to which the connection is reciprocal), and propinquity (degree to which individuals have more ties with people geographically close to them) [57].

***Patient journey mapping:*** Qualitative analysis will be conducted utilizing data collected from the gamified patient journey mapping exercise within the online survey. Data collected via researcher observations and field notes from the PSM process may also be used to further validate the data. Thematic analysis will be conducted to identify patterns or themes regarding whether participants’ contributions to the PSM process led to changes in perceived knowledge, beliefs, assumptions, or engagement behaviors with the local youth mental health system at each participating site.

***Prioritizing interventions in the youth mental health system:*** Data will be analyzed to understand the similarities and differences among participants across the system before and after participating in the PSM process (e.g., assessing convergence between respondents), as well as to understand and measure any change in participants’ beliefs of the priority of importance and funding allocations of youth mental health interventions (e.g., convergence with insights from the systems model to indicate systems learning).

***Prospective evaluation of impact using the systems models:*** The technical systems model that will be the final output of the PSM process will also provide analytic capacity. This will allow researchers and end users to simulate and track (and thus prospectively monitor and evaluate) both short- and long-term impacts, including but not limited to identifying unintended consequences through modelling as well as how suggested interventions of the systems model might impact outcomes such as death by suicide.

***Analysis of other sources of data:*** In addition to cross-validating data collected from the online surveys and interviews, qualitative data collected via researcher observations and recordings from the PSM workshops, meetings with local stakeholders outside of the workshops, reflections, and field notes will be analyzed as part of the ongoing reflection cycle enabled by PAR. Specifically, researchers will work collaboratively with the local site to engage in ongoing planning, action (iteration), reflection, and observation throughout the evaluation process with the aim of improving the methods and outcomes of PSM.

### 2.7. Data Storage and Security

All data will be securely stored in accordance with The University of Sydney’s data management procedures [38,39]. The importance of data sovereignty is recognized when working with Aboriginal and Torres Strait Islander people [40,41]. Where relevant, a data governance framework will be established through a consultative process at an early stage to allow for sufficient time to work with the participating site and their respective Aboriginal governance councils.

## 3. Discussion

Evaluations are crucial for determining whether the postulated benefits of PSM are realized and to inform refinements to the process as needed throughout all stages of the Program [26,58,59]. This novel study is the first to apply a comprehensive multi-scale evaluation framework. The feasibility, value, impact (change or action), and sustainability that is achieved among the participants as a result of the PSM process for the *Right care*, *first time*, *where you live* Program will be assessed.

A novel mixed methods approach is described to evaluate the PSM component of the Program. This approach facilitates participant engagement at various levels by providing different mechanisms for participants to contribute. Specifically, a mixed methods approach creates the opportunity for participants to contribute their more in-depth experiences through a 60-min interview and conversely creates the opportunity for participants to contribute to a less onerous 15-min online survey [60]. As it is more difficult to elicit in-depth thinking within online surveys compared with semi-structured interviews (i.e., where researchers have the opportunity to probe), the online survey questions have been purposefully designed to encourage engagement as well as in-depth participant thinking through gamification.

The incorporations of gamification, patient journey mapping, and social network analysis as part of the evaluation process are novel features. Though gamification of online surveys has been conducted in many disciplines [47,61], it has not yet been considered in PSM evaluation to our knowledge. Aside from increasing participant engagement, the approach of gamifying survey questions has many benefits, such as increasing motivation and enjoyment to complete an activity [61]. This approach can also empower participants, particularly more vulnerable groups, to share their experiences in more detail, facilitated by visual, non-text cues as opposed to just presenting participants with traditional survey methods such as multi-response questions [62].

The patient journey mapping activity, which is widely conducted to help understand how patients interact with health services or the broader social system throughout their care journey [63], will yield data that can be analyzed to observe changes to how participants viewed the local youth mental health systems (e.g., challenges, strengths, priorities, etc.) before and after the PSM process. In addition, data from the patient journey mapping activity can inform the PSM process to facilitate discussion points around system challenges and address any discrepancies during the PSM workshops, as well as inform the design of the systems model (e.g., contextualizing it in the local system context) [64].

Social network analysis has not previously been applied to understand the changes in interdisciplinary collaboration among participants as a result of the PSM process, despite the general acceptance that PSM can promote knowledge exchange and facilitate interdisciplinary collaboration between participants [28,32]. Toward this end, a visual map of the social network will be developed at three time points to measure how local social networks changed over the PSM process.

A PAR approach allows for the evaluation processes to be embedded within program planning, iteration, reflection, and observation. PAR aims to improve equitable health and social practice, enable authentic participation and collaboration, and empower individuals to improve their knowledge to facilitate measurable change [65]. Embedding PAR throughout the Program creates a culture of equality when engaging with diverse stakeholders to ensure that the local systems model is developed collaboratively, therefore increasing stakeholders’ assurance of the credibility of the systems model. PAR can also facilitate stakeholder learning, ensuring insights from the final decision support tool can be easily understood by diverse subgroups. This study also incorporates evaluation processes ex ante (before), ex durante (during), and ex post (after) to ensure that reflective evaluation processes are considered throughout all phases of the Program to improve the PSM process in the current research study, as well as to provide researchers with a flexible blueprint for implementation and replication in diverse contexts [59]. 

## 4. Ethics and Dissemination

The described evaluation study protocol has been reviewed and approved by the Sydney Local Health District Human Research Ethics Committee (HREC; Protocol No. X21-0151 and 2021/ETH00553). Any modifications to the study protocol will only be implemented after HREC approval.

The results of this evaluation study will be published in peer-reviewed journals and policy documents and may be presented at both domestic and international scientific meetings. The results may also be communicated directly to the wider local community, such as through public presentations and podcasts. All dissemination of findings will only report aggregate, non-identifiable data.

## 5. Conclusions and Future Directions

An innovative evaluation study is described, in which the combination of PSM and novel evaluation tools using gamification will be deployed for the first time in the complex decision-making environment of youth mental health systems. The knowledge generated by this evaluation study will uncover the social and technical opportunities arising from PSM processes (e.g., measuring individual, group, and system changes to understand the value of adopting a participatory approach when developing systems models). The evaluation study will also provide insight into the challenges of implementing these interdisciplinary methods in complex settings to improve youth mental health policy, planning, and outcomes.

Though described in the context of a youth mental health PSM program in Australia, the comprehensive multi-scale evaluation framework can be applied internationally across diverse disciplines and modelling methods. Adopting such an evaluation approach will likely lead to local capacity building, thus improving the methods to explore and address complex systems challenges through stakeholder understanding, consensus building, and collaborative action, all of which underpin genuine individual and community empowerment.

## Figures and Tables

**Figure 1 ijerph-19-04015-f001:**
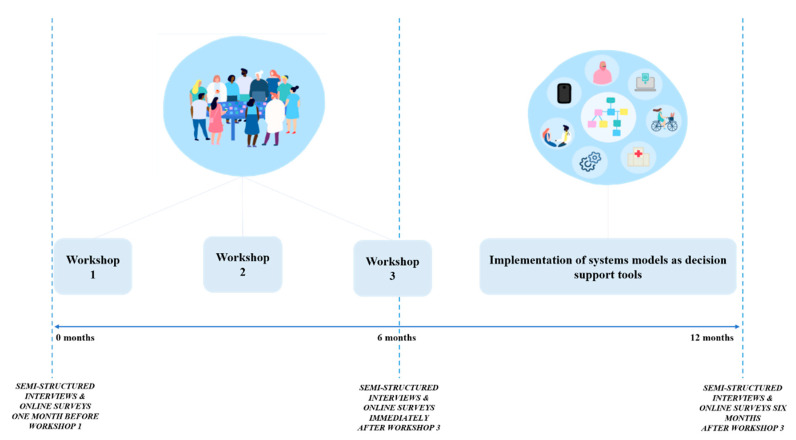
Evaluation time points aligned to the three PSM workshops. Evaluation will be conducted in two sites per year from 2022 to 2025 (total of eight sites).

**Figure 2 ijerph-19-04015-f002:**
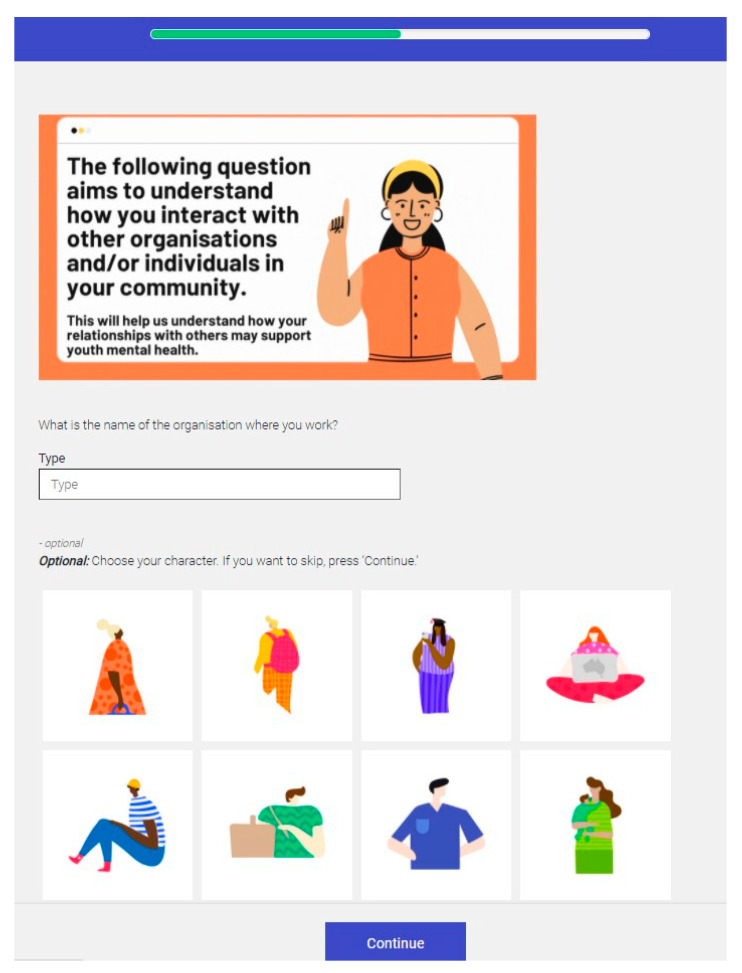
Preview of the gamified social network analysis activity to measure if interdisciplinary collaboration has changed (i.e., improved or been made worse) as a result of the PSM process.

**Figure 3 ijerph-19-04015-f003:**
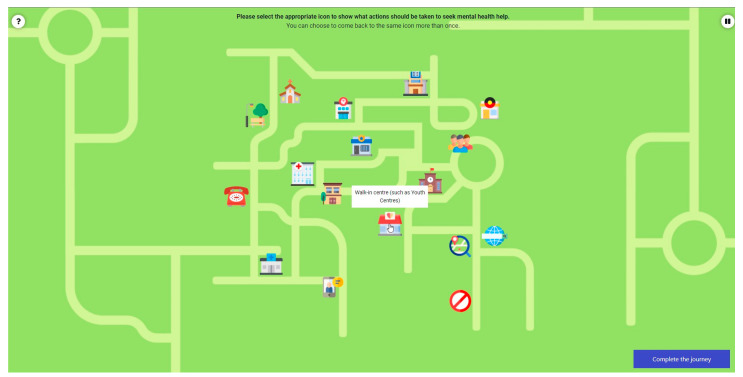
Preview of the patient journey mapping activity. Respondents will explore a fictional case study or will share their own personal experiences of how young people seek help and navigate their community’s youth mental health system.

**Table 1 ijerph-19-04015-t001:** Applying a comprehensive multi-scale evaluation framework to collect data utilizing a mixed-methods approach for the Program.

Evaluation Category	Evaluation Criteria or Question	Level of Impact	Data Collection Method				
			Online Surveys	Interviews	Workshop or Meeting Recordings	Field Notes or Other Documentation	Systems Model Tracking
**FEASIBILITY** *Is PSM feasible?*	Is it feasible to develop systems models through participatory methods for each participating site?	Project	X	X	X	X	X
	Is it feasible to recruit all necessary stakeholder perspectives in the PSM process?	Project	X	X	X	X	
	How do participants view the credibility of the PSM process?	Individual	X	X			
	How do participants contribute and engage during the PSM process?	Individual	X	X			
	How do participants view the credibility of the evidence used to effectively inform the systems model?	Individual					
	How were power relationships managed?	Group	X	X	X	X	
	Did all participants contribute and engage during the PSM process (e.g., inclusive, accessible, and transparent)?	Group	X	X	X	X	
	Can systems models be built through a participatory approach that can effectively inform policy, planning, and investment decisions with a degree of confidence in accuracy to improve youth mental health and wellbeing?	System	X	X			X
**VALUE** *What is the value of the PSM process?*	How did the PSM process add value (e.g., context, validity, learning, and salience) to developing the systems models?	Project	X	X	X	X	
	What are the facilitators and barriers to developing systems models through participatory methods (e.g., incentives, time, and resources)?	Project	X	X	X	X	
	What are the experiences (e.g., benefits and challenges) arising from the application of PSM (e.g., positive outcomes and ability to share personal stories)?	Individual	X	X			
	What are the experiences (e.g., benefits and challenges) of the participants using the local systems model decision support tool (e.g., confidence using the tool, ease or simplicity of use, and acceptance)?	Individual	X	X			X
	What are the experiences (e.g., benefits and challenges) working in interdisciplinary collaboration with diverse stakeholders for PSM (e.g., communication, relationships, trust, and social networks)?	Group	X	X			
	Does the participatory approach in building systems models add sufficient value to warrant the time and resources investment (e.g., improve capacity, efficiency, and confidence)?	System	X	X			
**CHANGE AND ACTION (IMPACT)** *What changed as a result of PSM?*	How was feedback considered throughout the program to improve the PSM process (including the build of the systems model)?	Project	X	X	X	X	
	Was the PSM process flexible enough to take action or respond to the changing needs of each of the eight participating site’s local youth mental health systems?	Project	X	X		X	
	Are there changes or what are the impacts in perceived knowledge, beliefs, behaviors, or assumptions?	Individual	X	X			X
	Are there changes, or what are the impacts in the way participants engage with their local youth mental health systems (e.g., reflection)?	Individual	X	X			X
	Are there changes in or what are the impacts of social network connections and interdisciplinary collaboration as a result of the PSM process?	Group	X	X			
	Are there changes or what are the impacts in perceived knowledge, beliefs, behaviors, or assumptions for broader stakeholders (e.g., organizational learning)?	Group	X	X			X
	How have insights from the PSM process been applied in the local youth mental health system?	System	X	X			X
	What are the factors that have influenced the extent to which the systems model has been utilized?	System	X	X	X	X	
**SUSTAINABILITY** *What are the outcomes of PSM over the longer term?*	How does the PSM process promote sustained use of the systems model?	Project	X	X			
	Are there sustained changes in knowledge, beliefs, behaviors, or assumptions for the participants (e.g., resilience to uncertainty)?	Individual	X	X	X		X
	Are changes in social network connections and interdisciplinary collaborations sustained over time?	Group	X	X			
	How have insights from the systems models been applied in the longer term?	System	X	X		X	
	How do participants’ engagement with and use of the systems model change over time?	System	X	X		X	X
	What are the longer term factors that have influenced the extent to which the systems model is ongoingly utilized to inform local youth mental health policy, planning, and investment decisions?	System	X	X		X	X

## Data Availability

Not applicable.

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
