# Peer review of "Participatory Systems Modelling for Youth Mental Health: An Evaluation Study Applying a Comprehensive Multi-Scale Framework"

_ijerph, 2022, doi:10.3390/ijerph19074015_

Round 1

Reviewer 1 Report

Please refer to the attached file

Author Response

Reviewer #1:

This study focuses on the topic of participatory systems modelling for youth mental health, highlighting novel data collection procedures, including online surveys combined with gamification, to enable social network analysis and patient journey mapping. The reviewer considered the topics and directions discussed in this study to be accepting. Nevertheless, it is recommended that the following questions be revised, as follows:

  1. What are the specific experimental results of this study? It is not displayed in the text, and the modeling results are not visible? Please show the results of the modeling!

  2. The sampling method using snowball sampling is a non-random sampling survey, and the results of the study cannot fully conform to scientific inferences and can only persuade some phenomena!

  3. Emphasizing "The evaluation design will be aligned to the PSM workshops and will be conducted in two sites per year from 2022 to 2025" in the 177 lines, if possible, can describe the period between 2022 and 2025 to assess the impact.

  4. In the 174 lines, can you specify the eight participating sites chosen and the conditions to be considered?

  5. The entire article gives the reader the impression that it is a draft research plan rather than a complete and practiced research report.

After further examination and modification, the paper can be considered for publication.

Our response: Many thanks for your thorough review. To respond to the reviewer questions:

  1. There are no specific experimental results of this study, as the submitted manuscript is a protocol paper. As mentioned in the reviewer’s fourth question, this study will be conducted from 2022 to 2025, so it is expected the first set of results will be submitted for publication in 2023. Importantly, the broader components of the Right care, first time, where you live research program are described and cited in the manuscript (lines 128 - 129).

  2. Thank you for your feedback. As stated in our manuscript (lines 213 – 217), the recruitment processes for this study will be guided by the primary partner organization at each of the participating sites, as community development and empowerment are fundamental to this research program. In instances where appropriate (line 215), active snowball sampling will be employed so that community members are given equal opportunity and empowered to identify other stakeholders to be invited to participate. In other words, active snowball sampling will not be the sole and primary method of recruitment.

  3. Thank you for your feedback. The methods of data collection are described in further sections of the manuscript. To provide the reader with the appropriate signpost, we have included in lines 179 – 180, “The evaluation approach, including the data collection procedures, are described below.”

  4. Unfortunately, we are unable to disclose the eight participating sites chosen as we have not yet executed the appropriate research collaboration agreements. However, we have included in lines 122 - 123 the regions where the sites will be located, “…in the Australian Capital Territory, New South Wales, Queensland, and Western Australia.”

  5. Thank you for your feedback. As stated throughout the manuscript, this paper is a study protocol. As this is a national Australian research study that is applicable for broader international implications, we hope to publish our protocol to increase transparency of our research approach, facilitating potential opportunities for further research collaboration.

Reviewer 2 Report

Thank you for the opportunity to review your article. I very much appreciated your introduction section, and I think you did a thorough job reviewing the current literature and the PSM process. I cannot tell you enough how appreciative I am that you all are including the youth themselves in your study. What an important stakeholder to include! I believe I was able to get a clear understanding of what will be included in this project through your explanations and figures. I am anxious to read about the results!

Strengths: --Very clear and understandable introduction section. --Article addresses the objectives that were set by the authors in an adequate manner --The population they will be conducting for the study discussed in this article is appropriate and representative of individuals that are important to include in examining the mental health context of Australia. --Enough information is provided by the authors to allow others to replicate similar studies across other contexts and countries.

Weaknesses: --I would like to see some explanation by the authors as to why they are choosing to only include age 18 and older youth with lived (or living with) mental health conditions and not youth younger than 18. The reason is because I think it is presented in the introduction as these youth being included as stakeholders in the evaluation study. I think their input is vital in understanding an aspect of mental health care and impact. However, some of these youth would be younger than 18. I know, at least in the United States, youth under 18 can participate with parental consent.

Author Response

Reviewer #2:

Thank you for the opportunity to review your article. I very much appreciated your introduction section, and I think you did a thorough job reviewing the current literature and the PSM process. I cannot tell you enough how appreciative I am that you all are including the youth themselves in your study. What an important stakeholder to include! I believe I was able to get a clear understanding of what will be included in this project through your explanations and figures. I am anxious to read about the results!

Strengths: --Very clear and understandable introduction section. --Article addresses the objectives that were set by the authors in an adequate manner --The population they will be conducting for the study discussed in this article is appropriate and representative of individuals that are important to include in examining the mental health context of Australia. --Enough information is provided by the authors to allow others to replicate similar studies across other contexts and countries.

Weaknesses: --I would like to see some explanation by the authors as to why they are choosing to only include age 18 and older youth with lived (or living with) mental health conditions and not youth younger than 18. The reason is because I think it is presented in the introduction as these youth being included as stakeholders in the evaluation study. I think their input is vital in understanding an aspect of mental health care and impact. However, some of these youth would be younger than 18. I know, at least in the United States, youth under 18 can participate with parental consent.

Our response: Thank you for your feedback. We look forward to sharing our study results in the future! We agree with your suggestion and have modified our inclusion criteria to also include individuals aged 14 – 17 years old (line 205). We have decided on this age range as we have received ethics approval to not seek consent parent/guardian consent for young people aged 14 to 17 years due to the following reasons:

  1. Research has identified that a request for written permission from parents can represent a threat to the validity of youth surveys relating to health and wellbeing (e.g. requiring parental consent can result in a significantly lower participation rate).[1] As a result, during the past 20 years, research groups from different countries have conducted youth surveys without parental consent.[2][3] In these studies, the view was upheld that older children are sufficiently mature to decide for themselves whether to participate or not.

  2. The UN Convention on the Rights of the Child views older children as persons with the right to joint determination. In a number of countries, the health and social systems accord young people this joint determination and express a regard for children’s and young people’s right to be heard on issues concerning them. Danish authorities, for example, had no objection to the conduct of a national survey among 15- to 16-year-olds provided that data could not be linked to any respondent.1 Furthermore, the Australian Government has stated that individuals aged 14 years and older can manage their own health record, without requiring oversight by parents or guardians.[4]

  3. The law in Australia recognizes this concept of the ‘mature minor’, which is founded in common law. The High Court of Australia considers that a child under the age of 18 years is capable of giving effective consent if they fully comprehend the nature, consequences and risks of the proposed action, irrespective of whether a parent consents.

[1] Helweg-Larsen, K., & Bøving-Larsen, H. (2003). Ethical Issues in Youth Surveys: Potentials for Conducting a National Questionnaire Study on Adolescent School Children’s Sexual Experiences with adults. American Journal of Public Health, 93(11), 1878-1882.

[2][2] Edgardh, K., & Ormstad, K. (2000). Prevalence and characteristics of sexual abuse in a national sample of Swedish seventeen‐year‐old boys and girls. Acta paediatrica, 89(3), 310-319.

[3] Sariola, H., & Uutela, A. (1996). The prevalence and context of incest abuse in Finland. Child abuse & neglect, 20(9), 843-850.

[4] Australian Digital Health Agency (2019). Manage your record from age 14. Accessed May 2019. https://www.myhealthrecord.gov.au/for-you-your-family/howtos/manage-your-record-from-age-14  

Reviewer 3 Report

It is an excellent work with a good foundation and a rigorous methodology to carry it out, congratulations. Just a small specification:

citation 7 (line 46). It refers to an article published in Lancet in 2011. It is not appropriate to use such old references for data that change rapidly, as is the case here.

It is recommended to clearly specify the exclusion criteria of the sample.

Author Response

Reviewer #3:

It is an excellent work with a good foundation and a rigorous methodology to carry it out, congratulations. Just a small specification: citation 7 (line 46). It refers to an article published in Lancet in 2011. It is not appropriate to use such old references for data that change rapidly, as is the case here.

Our response: Thank you for your feedback. We have updated our reference to include a more recent publication. We intentionally did not explicitly state our exclusion criteria as we have clearly indicated our inclusion criteria (and thus felt that an exclusion criteria would be redundant).